# Lot quality assurance sampling survey for water, sanitation and hygiene monitoring and evidence-based advocacy in Bentiu IDP camp, South Sudan

**Berhe Etsay Tesfay**[1¤a]*, **Destaw Gobezie**[1], **Ivan Andreas Sinaga**[1], **Amanya Jacob**[2], **Abdul Wasay Mullahzada**[3], **Samreen Hussain**[1¤b], **Rosita de Boer**[1], **Biserka Pop-stefanija**[3], **Monika Slosarska**[1], **Patrick Keating**[4]

1 Médecins Sans Frontières, Juba, South Sudan, 2 Ministry of Health, Juba, Republic of South Sudan, 3 Public Health Department, Médecins Sans Frontières, Amsterdam, Netherlands, 4 Public Health Department, Médecins Sans Frontières, London, United Kingdom

¤a Current address: Liverpool School of Tropical Medicine, Liverpool, United Kingdom
¤b Current address: Public Health Department, Médecins Sans Frontières, Amsterdam, Netherlands
* berhe.etsay@gmail.com

**Data Availability Statement:** Raw data cannot be shared publicly because the data includes sector and block information, which could be sensitive

## Abstract

### Background

Every year, 60% of deaths from diarrhoeal disease occur in low and middle-income countries due to inadequate water, sanitation, and hygiene. In these countries, diarrhoeal diseases are the second leading cause of death in children under five, excluding neonatal deaths. The approximately 100,000 people residing in the Bentiu Internally Displaced Population (IDP) camp in South Sudan have previously experienced water, sanitation, and hygiene outbreaks, including an ongoing Hepatitis E outbreak in 2021. This study aimed to assess the gaps in Water, Sanitation, and Hygiene (WASH), prioritise areas for intervention, and advocate for the improvement of WASH services based on the findings.

### Methods

A cross-sectional lot quality assurance sampling (LQAS) survey was conducted in ninety-five households to collect data on water, sanitation, and hygiene (WASH) coverage performance across five sectors. Nineteen households were allocated to each sector, referred to as supervision areas in LQAS surveys. Probability proportional to size sampling was used to determine the number of households to sample in each sector block selected using a geographic positioning system. One adult respondent, familiar with the household, was chosen to answer WASH-related questions, and one child under the age of five was selected through a lottery method to assess the prevalence of WASH-related disease morbidities in the previous two weeks. The data were collected using the KoBoCollect mobile application. Data analysis was conducted using R statistical software and a generic LQAS Excel analyser. Crude values, weighted averages, and 95% confidence intervals were calculated for

and potentially pose risks if people were able to track back to specific blocks. Data are available from oca.research@london.msf.org (Head of research and ethics committee) and oca.data@london.msf.org (Data manager) for researchers who meet the criteria for access to confidential data.

**Funding:** The author(s) received no specific funding for this work.

**Competing interests:** The authors have declared that no competing interests exist.

each indicator. Target coverage benchmarks set by program managers and WASH guidelines were used to classify the performance of each indicator.

## Results

The LQAS survey revealed that five out of 13 clean water supply indicators, eight out of 10 hygiene and sanitation indicators, and two out of four health indicators did not meet the target coverage. Regarding the clean water supply indicators, 68.9% (95% CI 60.8%-77.1%) of households reported having water available six days a week, while 37% (95% CI 27%-46%) had water containers in adequate condition. For the hygiene and sanitation indicators, 17.9% (95% CI 10.9%-24.8%) of households had handwashing points in their living area, 66.8% (95% CI 49%-84.6%) had their own jug for cleansing after defaecation, and 26.4% (95% CI 17.4%-35.3%) of households had one piece of soap. More than 40% of households wash dead bodies at funerals and wash their hands in a shared bowl. Households with sanitary facilities at an acceptable level were 22.8% (95% CI 15.6%-30.1%), while 13.2% (95% CI 6.6%-19.9%) of households had functioning handwashing points at the latrines. Over the previous two weeks, 57.9% (95% CI 49.6–69.7%) of households reported no diarrhoea, and 71.3% (95% CI 62.1%-80.6%) reported no eye infections among children under five.

## Conclusion

The camp's hygiene and sanitation situation necessitated immediate intervention to halt the hepatitis E outbreak and prevent further WASH-related outbreaks and health issues. The LQAS findings were employed to advocate for interventions addressing the WASH gaps, resulting in WASH and health actors stepping in.

## Background

The United Nations adopted the Human Right to Safe Drinking Water and Sanitation (HRTWS), which calls for universal access to safe, affordable, acceptable, available, and accessible water, sanitation and hygiene (WASH) services by 2030 [1]. WASH initiatives are crucial in reducing poverty, promoting equality, and supporting socioeconomic development [2]. These services were targets under the Millennium Development Goals (MDGs) for 2015 and are now part of the Sustainable Development Goals (SDGs) for the post-2015 period [3]. However, vulnerable populations, particularly in developing countries and refugee settings, have limited access to WASH services, impacting individuals' and societies' health and social well-being [4]. According to the United Nations (UN), nearly half of the global population lacks safe sanitation, and over 2.2 billion people do not have access to safe drinking water. Approximately 2 million individuals worldwide do not have access to handwashing facilities with soap, and half a million still practice open defecation [5].

Globally, approximately 88% of deaths due to diarrhoeal diseases are attributed to inadequate safe water and poor hygiene and sanitation, of which 60% occur in low- and middle-income countries (LMIC) [6, 7]. These diarrhoeal diseases (including cholera) kill more children than AIDS, malaria, and measles combined, making diarrhoeal diseases the second leading cause of infectious disease death after pneumonia among children under five, excluding neonatal death [7, 8]. In 2016, inadequate WASH contributed to 60% of diarrhoeal deaths,

which could have been prevented by improving water and sanitation services [9]. Furthermore, improved hygiene, sanitation, and safe water access can reduce neglected tropical diseases (such as schistosomiasis and Guinea worm diseases) morbidities by almost 80% [10]. Access to safe water, improved hygiene, and sanitation have the potential to prevent at least 9.1% of the global disease burden and 6.3% of all global deaths [11].

Populations in displacement camps and refugee settings are usually in precarious situations and particularly prone to hygiene and sanitation-related diseases. Bentiu Internally Displaced Population Camp (IDP) in South Sudan was established in 2013 as a protection of civilians' camp under the UN for people who sought safety and protection, following the civil war between government and opposition forces. The camp is home to approximately one hundred thousand people, which varies slightly between different seasons (e.g. due to flooding) and ongoing conflicts in the surrounding area [12, 13]. The nature of the camp presents distinct health challenges due to ongoing insecurity, population movement, seasonal weather variation, climate impact, food insecurity, lack of sufficient water, lack of sufficient hygiene and sanitation services, and the presence of endemic infectious diseases. Primary health risks include diarrhoeal disease (acute watery and acute bloody diarrhoea, including cholera), hepatitis E infection, seasonal malaria, and other vector and waterborne illnesses.

The provision of WASH services in the Bentiu IDP camp was shared among several WASH actors. Those WASH actors are heavily dependent on external funding. Due to the coronavirus disease (COVID-19) pandemic, there was a shift in the budget and effort towards COVID-19 treatment and prevention activities. As primarily a health actor, Médecins Sans Frontières (MSF) is not responsible for the WASH activities in Bentiu IDP camp but responds to the high levels of WASH-related diseases faced by the community. In addition, MSF is a direct witness of the impact of WASH gaps on the community's health and is thus usually in a good position to advocate for appropriate WASH interventions.

Different organisations and institutions have used different methods to assess the WASH situations in camp settings. The United Nations Humanitarian Charter for Refugees (UNHCR), WASH in emergency handbook and Sphere guidelines, annual knowledge, attitude, and practice (KAP) surveys, and monthly reports are mentioned as means of assessment, monitoring, and evaluation [14]. Lot quality assurance sampling (LQAS) is an alternative approach for assessing the WASH situation in camp settings [15]. The method has been used to assess a variety of health services, including vaccination coverage, disease prevalence and health care monitoring [16–18]. This method has shown valid outputs at a lower cost and is quicker and easier to implement than extensive surveys [16–19]. Moreover, the findings were used for programmatic monitoring, evaluation, and improvement. This study used the LQAS methodology to quantify the WASH gaps in the Bentiu IDP camp, identify priority intervention areas, mobilise action resources, and support advocacy efforts.

## Methodology

### Study setting and population

Bentiu IDP camp is located in Unity State, South Sudan. It hosts a population of approximately 107,000 people, of which 50.6% are female. The camp is divided into five sectors, which are further divided into 64 blocks, housing from approximately 1,000 individuals to more than 3,200 individuals each [13]. The target population comprised all households in all five sectors of the Bentiu IDP camp.

## Study design

A cross-sectional lot quality assurance sampling (LQAS) survey was conducted from August 2nd to 6th, 2021, in Bentiu IDP camp, South Sudan. This survey methodology uses small sample sizes and allows the classification and prioritisation of needs on a smaller geographic level management unit (called the supervision area [SA]) [20].

## Study sample

A sample size is established at the SA level, and a decision rule is selected (which depends on the sample size), which is the cut-off below which an area is classified as low-performing for an indicator [20]. Nineteen households are typically sampled per SA, which ensures that the α (probability of misclassifying an area with high coverage as low) and β (probability of misclassifying an area with low coverage as high) errors are both maintained at 10% (S1 Table) [21].

In this survey, two client populations or 'universes were sampled. The first client population was made up of households in the entire camp, and the second client population of parents/guardians of children aged less than five years in the households selected for the first population. Nineteen households were sampled from each SA, giving a sample size of 95 for each client population. Probability proportional to size (PPS) sampling was used to identify the blocks from which the 19 households in each SA were selected. Households were selected using random sampling using a Geographic Positioning System (GPS) in Quantum Geographic Information System (QGIS). The selected households were imported into the OpenStreetMap Automated Navigation Directions (OsmAnd) map mobile application to navigate to the household during data collection. A person older than 18 years and with more information in the household or who had spent more time in the household was selected and requested to be the respondent. One child under five from the same household was included to assess the prevalence of WASH-related diseases over the previous two weeks. For households with more than one child under five, a lottery method was used to select one of them.

## Data collection

Two questionnaires (one for each target client population) were used to collect data. One set of questionnaires was about WASH-related indicators, and the other set was about the prevalence of WASH-related disease morbidities in children under five, based on previously used and tested indicators and question sets (S3 File). The data collectors were trained for three days, and a pilot test was conducted for one day. Data were collected using KoBoCollect (https://kobo.msf.org) using smartphones [22].

## Data analysis

Data were imported into R software version 4.1.2 for descriptive statistical analysis, and a generic Microsoft Excel LQAS analyser from the Liverpool School of Tropical Medicine (LSTM) was used to identify the priorities based on the decision rule table [23]. To determine the prevalence and coverage of the indicators at a camp-wide level, responses from all 19 respondents in all five SAs were combined for a total sample size of 95. This allowed for the calculation of the crude average coverage or prevalence for the entire IDP camp. Using population data from each supervision area, a weighted average coverage or prevalence was calculated for the entire IDP camp and each supervision area.

Responses were calculated and compared to the decision rules per indicator and the crude and weighted averages using the LQAS table. Data from all SAs were aggregated, and coverage

indicators for the whole camp were determined based on indicators from the LQAS Generic Health Results Excel Sheet [24].

The target coverage (%) and target coverage DR (n) are established performance benchmarks for coverage per indicator set by program managers in previous LQAS surveys and targets available in the UNHCR, United States Agency for International Development (USAID), and Sphere Guidelines to rule if the indicator has met the target. First, the aggregated weighted average coverage for an indicator for the entire camp was calculated and compared with the LQAS table (S1 Table). For example, if the weighted average was 70% for a specific indicator, this equated to a score of 11 out of 19 in the LQAS table (S1 Table). This is the average DR(n) referred to in this study. The target and Average DRs were then used to classify each indicator's high-priority, medium-priority, and low-priority areas. If the SA's specific indicator performance was less than the target DR and less than the average DR, the SA was classified as a high priority. If the SA's specific indicator performance was above the average DR but less than the target DR, it was classified as medium priority. If the SA's performance was higher than or equal to the target DR, it was classified as low/no priority. In addition, the performance of indicators from this LQAS survey was compared to the most recent LQAS survey (2019).

## Ethics

This study was conducted in accordance with the World Medical Association Declaration of Helsinki Ethical Principles for Medical Research Involving Human Subjects (2013) and the 2016 International Ethical Guidelines for Health-Related Research Involving Human Subjects (CIOMS) [25, 26]. Ethical approval was obtained from the MSF Ethical Review Board and the Ministry of Health South Sudan Research Ethical Review Board. Community and stakeholder engagement was conducted before the study through camp management and meetings with community leaders. The data collectors administered an information sheet and provided the participants with a copy of the translated information sheet. Verbal informed consent was obtained from all participants.

## Results

The survey included 95 household representatives and 95 children under five. The average household size was eight people. Most respondents (81%) were females (Table 1).

### Water supply indicators prevalence and coverage

Eight out of sixteen indicators of clean water supply met the target coverage. Most households 95.0% (95% CI 90.5%-99.5%) reported that they get water for drinking, washing hands, and dishes from the potable water source or tap stand during the dry and rainy seasons, while 46% (95% CI 36%-57%) of the households had at least 15 litres of water per person per day. In addition, 68.9% (95% CI 60.8%-77.1%) of the households reported water available from the source at least six days a week, and 51.4% (95% CI 44.1–58.7%) of the households reported that they filled their water containers before the water tap stopped running. 36.6% (95% CI 26.4–46.8%) of the households can store drinking water for less than a day. Households with acceptable water storage containers (at least narrow mouth, clean, and have a lid) accounted for 36.8% (95% CI 26.5–46.9%) (Table 2).

### Coverage of hygiene indicators

None of the six hygiene indicators met the target coverage. The proportion of households that had their water jugs for cleansing after defaecation was 66.8% (95% CI 49–84.6%), and 17.9% (95% CI 10.9–24.8%) of the households had handwashing setups within their living area. A

**Table 1. Socio-demographic characteristics of the respondents and the households included in the Bentiu IDP WASH survey, August 2021.**

| Variable | Overall, N = 95[1] | SA 1, N = 19[1] | SA 2, N = 19[1] | SA 3, N = 19[1] | SA 4, N = 19[1] | SA 5, N = 19[1] |
|---|---|---|---|---|---|---|
| **Household size** | 7 (6, 9) | 9 (7, 10) | 7 (5.5, 9.5) | 7 (6, 9) | 8 (6.5, 8.5) | 7 (5, 10) |
| **Female head household** | 82 (86%) | 18 (95%) | 13 (68%) | 17 (89%) | 19 (100%) | 15 (79%) |
| **Sex of respondent** | | | | | | |
| Female | 77 (81%) | 11 (58%) | 15 (79%) | 17 (89%) | 19 (100%) | 15 (79%) |
| Male | 18 (19%) | 8 (42%) | 4 (21%) | 2 (11%) | 0 (0%) | 4 (21%) |
| **People 50 years and older in the house** | | | | | | |
| 0 | 25 (26%) | 0 (0%) | 5 (26%) | 3 (16%) | 8 (42%) | 9 (47%) |
| 1 | 54 (57%) | 11 (58%) | 13 (68%) | 13 (68%) | 8 (42%) | 9 (47%) |
| 2 | 16 (17%) | 8 (42%) | 1 (5.3%) | 3 (16%) | 3 (16%) | 1 (5.3%) |
| **Number of under five children** | | | | | | |
| two or less | 69 (73%) | 10 (53%) | 16 (84%) | 16 (84%) | 13 (68%) | 14 (74%) |
| 2–4 | 25 (26%) | 8 (42%) | 3 (16%) | 3 (16%) | 6 (32%) | 5 (26%) |
| 4 and above | 1 (1.1%) | 1 (5.3%) | 0 (0%) | 0 (0%) | 0 (0%) | 0 (0%) |
| **Number of under-five children and older than 50 years old** | | | | | | |
| two or less | 38 (40%) | 1 (5.3%) | 10 (53%) | 8 (42%) | 8 (42%) | 11 (58%) |
| 2–4 | 47 (49%) | 11 (58%) | 9 (47%) | 9 (47%) | 11 (58%) | 7 (37%) |
| 4 and above | 10 (11%) | 7 (37%) | 0 (0%) | 2 (11%) | 0 (0%) | 1 (5.3%) |

[1]Median (IQR); n (%)

quarter, 26.4% (95% CI 17.4–35.3%) of households could show one piece of soap, while 57.6% (95% CI 48.4–66.7%) of the households did not wash a dead body and did not wash their hands in a shared bowl after a funeral. As is common in Nuer culture, nearly all households reported eating from a shared plate (Table 3) [27].

## Coverage of sanitation indicators

Two of the four sanitation indicators met the threshold for target coverage. While 90.5% (95% CI 84.7–96.4%) of households reported using a pit latrine facility for toilet purposes, during an observation, 22.8% (95% 15.6–30.1%) of the households had a sanitation facility at an acceptable level (i.e. it at least had a door, was not full, and the slab was not falling). Almost all (97.4%, 95% CI 94.5–100%) menstruating women in the households reported that they used acceptable dignity kits (either sanitary pads or disposable/reusable cloth), which slightly exceeded the specified target (Table 4).

## Health indicators

Almost two-thirds (59.7%, 95% CI 49.6–69.7%) of parents/guardians reported that their under-five children did not have diarrhoea in the two weeks preceding the survey. In addition, 71.3% (95% CI 62.1–80.6%) of the parents reported no eye infections in the children under five in the preceding two weeks (Table 5).

## Comparison to the previous LQAS surveys in Bentiu IDP camp (Monitoring and evaluation)

The last LQAS survey in Bentiu IDP camp was conducted in 2019 (S1 File). Hence, we sought to compare the results from that survey to the most recent similar survey to identify whether the WASH situation in the camp had improved or deteriorated.

**Table 2. Water supply and quality coverage indicators in Bentiu IDP camp, August 2021.**

| Indicator | SA1 | SA2 | SA3 | SA4 | SA5 | Weighted average | Average DR†† | Target coverage† | Target DR† |
|---|---|---|---|---|---|---|---|---|---|
| Proportion of households that report using a potable water source for drinking both in dry and rainy season | 17 | 19 | 18 | 18 | 18 | 95% (90.5–99.5) | 16 | 95% | 16 |
| Proportion of households that report using PUR or AQUATAB sachets to treat rainwater | 0 | 0 | 0 | 0 | 0 | N/A | N/A | 95% | 16 |
| Proportion of households that report that water was available from their water source at least six of the seven days | 12* | 14* | 15* | 4** | 17 | 68.9% (60.8–77.1) | 11 | 95% | 16 |
| Proportion of households that report that they always get their containers filled from the tap stand before the water will stop running | 15 | 16 | 7** | 11 | 5** | 51.4% (44.1–77.1) | 8 | 95% | 16 |
| Proportion of household that find the taste of the water from the tap stand acceptable | 15 | 11* | 11* | 15 | 12 | 65.5% (55.6–75.3) | 10 | 75% | 12 |
| Proportion of households that report using a potable water source for cooking | 18 | 19 | 19 | 19 | 19 | 99.3% (NA) | N/A | 95% | 16 |
| Proportion of households that report using a potable water source for washing dishes | 19 | 19 | 19 | 19 | 19 | 100% (NA) | N/A | 95% | 16 |
| Proportion of households that report using a potable water source for washing your hands | 19 | 19 | 19 | 19 | 19 | 100% (NA) | N/A | 95% | 16 |
| Proportion of households that report using a potable water source for washing their clothes | 19 | 17 | 13 | 12 | 9** | 69.8% (60.6–79.0) | 11 | 65% | 10 |
| Proportion of households that report using a potable water source for bathing | 19 | 19 | 17 | 19 | 17 | 94.4% (89.3–99.6) | N/A | 80% | 13 |
| Proportion of households that have at least one water container that can hold water | 17 | 19 | 19 | 17 | 18 | 95.5% (91.7–99.4) | 16 | 95% | 16 |
| Proportion of households that had at least 40L of water the day before | 4** | 19 | 18 | 19 | 14 | 81.9% (75.7–88.1) | 14 | 95% | 16 |
| Proportion of households with acceptable water storage containers (at least narrow mouth, clean and has a lid) | 4** | 11 | 5** | 6* | 6* | 36.8% (26.5–46.9%) | 5 | 95% | 16 |
| Proportion of households that keep water in containers for less than one day | 3** | 12 | 6* | 5** | 6 | 36.6% (26.4–46.8) | 5 | 95% | 16 |
| Households with at least 15L of water per person per day | 5 | 9 | 10 | 10 | 8 | 46% (36–57%) | 7 | 95% | 16 |
| Water from the tap stand available at any time in the day | 0 | 0 | 0 | 0 | 0 | N/A | N/A | N/A | N/A |

† Established performance benchmarks for coverage per indicator set by program managers

†† LQAS value for the Weighed average (this is the cut-off point for medium versus high priority)

** High priority at camp level (indicators didn't meet the target for the average coverage of the SAs')

* Medium priority (indicators that didn't meet the target coverage set by program managers and guidelines)

All water supply indicators in the 2021 survey remained comparable to the findings of the 2019 LQAS survey and met the target coverage set by program managers and guidelines. However, most hygiene and sanitation indicators had deteriorated compared to the findings of the same survey conducted in 2019. Soap availability per household, hygiene promotion activities, and use of improved sanitation facilities all deteriorated compared to the findings from the 2019 survey. All other indicators remained comparable to the findings in 2019. The prevalence of hygiene-related diseases has also increased, but it is comparable to that found in the LQAS survey 2019 (S1 File).

## Discussion

The survey provided an overview of the WASH situation in Bentiu IDP camp. Fifteen out of twenty-seven indicators did not meet the target coverage levels set by the program managers. Hygiene and sanitation indicators showed the highest gap compared to the expected targets/benchmarks.

**Table 3. Hygiene practice and coverage indicators, Bentiu IDP camp, August 2021.**

| Indicator | SA1 | SA2 | SA3 | SA4 | SA5 | Weighted average | Average DR | Target coverage† | Target DR† |
|---|---|---|---|---|---|---|---|---|---|
| Proportion of households that report having their own water jug for cleansing after defecation | 15 | 15 | 14* | 9** | 11 | 66.8% (49–84.6%) | 11 | 95% | 15 |
| Proportion of households that have a hand washing area within their living area | 10* | 6** | 3** | 0** | 1** | 17.9% (10.9–24.8%) | 1 | 95% | 11 |
| Proportion of households that can show at least one piece of soap | 6* | 4* | 3* | 4* | 8* | 26.4% (17.4–35.3%) | 3 | 95% | 16 |
| Proportion of households that have been visited by a hygiene promoter within the last week | 6** | 10** | 15* | 7** | 6** | 49% (39.6–58.5%) | 7 | 95% | 16 |
| Proportion of households that do NOT eat from a shared plate | 1** | 1** | 0** | 1** | 3** | 6.5% (1.5–11.6%) | N/A | 75% | 10 |
| Proportion of households that do NOT wash a dead body AND do NOT wash hands in a shared bowl at a funeral | 16 | 13* | 4** | 13** | 13** | 57.6% (48.4–66.7%) | 9 | 95% | 16 |

† Established performance benchmarks for coverage per indicator set by program managers

†† LQAS value for the Weighed average (this is the cut-off point for medium versus high priority)

** High priority at camp level (indicators didn't meet the target for the average coverage)

*Medium priority (indicators that didn't meet the target coverage set by program managers and guidelines)

The UNHCR and MSF targets for clean drinking and cooking water were met in the survey. However, only 68.9% (95% CI 60.8–77.1%) of the households had access to water for at least six out of seven days, and only half of the households could get their containers filled before the water tap stopped running. According to the minimum standards for Sphere and UNHCR guidelines, water should be accessible at least eight hours daily in an area with 250 people per tap stand that flows 7.5 litres of water/ minute or 17 litres/minute for 500 people [22, 23]. This standard is not met in the camp, where 98% of respondents reported that water was available only a maximum of two times a day based on the schedule. In addition, only 46% of households met the minimum water availability per household (15 L of water per person per day in an emergency and 20L/per person/day in a post-emergency setup), which indicated that the water supply in the IDP camp was not sufficient to meet the population's needs [14]. The proportion of households that used tap water to wash their clothes was 69.8% (95% CI 60.6–79.0%). Water from the oxidation point (where the sewage of the whole camp is collected) is

**Table 4. Sanitation indicators practice and coverage in the Bentiu IDP camp in August 2021.**

| Indicator | SA1 | SA2 | SA3 | SA 4 | SA5 | Weighted average | Average DR | Target coverage† | Target DR† |
|---|---|---|---|---|---|---|---|---|---|
| Proportion of households that report using an improved sanitation facility | 16 | 19 | 15 | 17 | 19 | 90.5% (84.7–96.4%) | 16 | 95% | 16 |
| Proportion of households whose sanitation facility is observed to be in an acceptable condition | 1** | 3* | 2* | 14* | 3* | 22.8% (15.6–30.1%) | 2 | 90% | 15 |
| Proportion of households that have an acceptable hand washing area by the toilet facility they use† | 0** | 5** | 6** | 0** | 0** | 13.2% (6.6–19.9%) | N/A | 90% | 15 |
| Proportion of households whose female members use acceptable materials for menstrual hygiene | 18 | 16 | 19 | 19 | 19 | 97.4% (94.5–100%) | N/A | 95% | 16 |

† Established performance benchmarks for coverage per indicator set by program managers

†† LQAS value for the Weighed average (this is the cut-off point for medium versus high priority)

** High priority at camp level (indicators didn't meet the target for the average coverage)

*Medium priority (indicators that didn't meet the target coverage set by program managers and guidelines)

**Table 5. Prevalence of water, sanitation, and hygiene-related disease indicators in the Bentiu IDP camp, August 2021.**

| Indicator | SA1 | SA2 | SA3 | SA4 | SA5 | Weighted average | Average DR | Target coverage† | Target DR† |
|---|---|---|---|---|---|---|---|---|---|
| Proportion of parents/guardians who report NOT having diarrhea among children <5 years in last two weeks | 12* | 13* | 13* | 10* | 9* | 59.7% (49.6–69.7%) | 9 | 90% | 15 |
| Proportion of parents/guardians who report NOT having eye infection among children <5 years in last two weeks | 16 | 13* | 14* | 17 | 10 | 71.3% (62.1–80.6%) | 12 | 90% | 15 |
| Proportion of parents/guardians who report NOT having ear infection among children <5 years in last two weeks | 19 | 19 | 15 | 17 | 18 | 91.2% (85.1–97.2%) | 16 | 90% | 15 |
| Proportion of parents/guardians who report NOT having skin infection among children <5 years in last two weeks | 18 | 19 | 14* | 16 | 19 | 89.5% (83.9–95.2%) | 15 | 90% | 15 |

† Established performance benchmarks for coverage per indicator set by program managers

†† LQAS value for the Weighed average (this is the cut point for medium versus high priority)

** High priority at camp level (indicators didn't meet the target for the average coverage)

*Medium priority (indicators that didn't meet the target coverage set by program managers and guidelines

also used to wash clothes, a potential source of all kinds of waterborne diseases, including hepatitis E, cholera, typhoid, and others. This was in line with the concurrent outbreak of the hepatitis E virus in the Bentiu IDP camp and an increase in diarrhoeal diseases. There was a large gap between the number of latrines in an acceptable condition compared to the number of inhabitants in the camp, with more than 70 people per latrine. According to the Sphere guideline, the maximum is 20 people/latrine [28]. In addition, as unsafe latrines are risky for women and girls, poorly maintained latrines are less used by communities where women and girls prefer to go to the bush [28].

There might be many reasons to explain the continued poor hygiene and sanitation or deterioration of some of the services compared with the findings of the 2019 LQAS survey. The impact of COVID-19 was also one of the major contributing factors. While most of the WASH and health actors and funds were focusing on the prevention and management of COVID-19, there was also a budgeting and fund shift from the WASH sector to COVID-19 treatment and prevention activities. In addition, due to COVID-19, mass community mobilisation and hygiene mobilisation activities were restricted or limited to small numbers. Furthermore, recommendations from the 2019 LQAS survey were not advocated and enacted thoroughly because of the emergence of the COVID-19 pandemic and deprioritisation of related activities. In addition, there was a major fund cut starting at the beginning of 2021 due to the government declaration of the transition of the camp from protection of civilian camp to IDP. The population number in the IDP camp is also dynamic, with regular population influxes due to conflicts and natural disasters, according to the IOM displacement tracking tool [13]. This increased the demand for additional WASH facilities and may have overloaded existing ones, which may have directly impacted the WASH conditions in the camp.

In this survey, 40% of the children experienced at least one waterborne disease, mainly diarrhoea, followed by eye infection within the two weeks preceding the survey. This is in parallel with the deteriorated hygiene and sanitation situation in the IDP camp as a health consequence, including the hepatitis E virus outbreak occurring in the camp in 2021. In the whole year of 2021, there was also an increase in the number of diarrhoeal diseases in the outpatient department of the MSF hospital. This increase was higher than the seasonal diarrhoeal diseases increase in previous years. There was a tenfold increase in July 2021 compared to the average number of cases in April-June the same year. In addition, the number of cases with acute jaundice syndrome doubled since April 2021, with a total of 469 RDT-positive hepatitis E cases in

Bentiu MSF hospital and five deaths, including one pregnant mother, in 2021. Finally, there were 415 diarrhoeal disease cases, associated admissions, and 15 deaths in Bentiu Hospital in 2021.

While the LQAS survey was implemented to evaluate the current levels of the WASH gaps, this has also proved that we can use the findings for quicker and more effective advocacy and that an LQAS can be more than just a monitoring and evaluation tool.

We conducted extensive communication and advocacy on the identified gaps, and this gained attention from international donors, national health, and WASH actors, resulting in a physical visit to the site and, ultimately, funding to support activities in the camp. Consequently, the various identified gaps were enacted by the different actors in the camp, and funds were secured from donors. Activities that included the construction of hundreds of replacement latrines, managing dry waste, and establishing a faecal waste management centre were commenced immediately by different actors.

Three weeks after the survey, unexpected flooding occurred in the area, which affected the camp directly and through an influx of additional population displaced from the surrounding area. The survey results helped to justify the additional burden in the camp and advocate for an upgrade to the water and sanitation response to an emergency level, where MSF and other WASH actors initiated interventions later.

This study had a few limitations. First, it focused mainly on the coverage of the WASH situation in the Bentiu IDP camp. However, it did not assess knowledge, attitude, and utilisation practices, except via a small number of indicators. Access to services without the knowledge of service utilisation limits the full picture of the WASH services in the camp. In addition, most of the questions were based on the self-report of the household representative for the interview. This might have introduced social desirability bias. In addition, this survey did not include a qualitative approach (i.e. focus group discussions, key informant interviews, etc.), which could have provided a more in-depth explanation for some of the hygiene and sanitation practices reported by the community. Moreover, the small sample sizes of the LQAS surveys indicate that they are not highly powered, and the $\alpha$ and $\beta$ errors of 10% are still relatively high. This can result in incorrect classification of poorly performing areas as having higher or acceptable performance. As a result, it provides a wider confidence interval, and making an inference from the confidence intervals is less precise. While acknowledging the limitations of LQAS, we can also highlight our trust in the findings due to 1) consistency of findings across time with previous surveys and 2) in line with the health facility surveillance data of high AWD and hepatitis E.

In conclusion, the LQAS survey identified key gaps in WASH service provision to the internally displaced Bentiu population that needed a rapid response to mitigate and prevent further WASH-related outbreaks. Moreover, this study highlighted that LQAS surveys, combined with a strong advocacy component, can be used for quick evidence synthesis to inform timely interventions for WASH-related gaps in humanitarian camp settings.

## Supporting information

**S1 File. Comparison of the results to the 2019 LQAS survey results.**
(DOCX)

**S2 File. List of water hygiene and sanitation indicators.**
(DOCX)

**S3 File. The questionnaire used in this WASH study.**
(DOCX)

**S4 File. Sector-wise (Supervision area) level descriptive analysis.**
(DOCX)

**S1 Table. Generic LQAS table for decision rules of target and average coverage.**
(DOCX)

## Acknowledgments

Our gratitude goes to the Ministry of Health, South Sudan for their collaboration in conducting this survey. We acknowledge the people and camp management of the Bentiu IDP camp for tireless collaboration during the planning and data collection of this survey and collaboration during the action plan for the intervention.

## Author Contributions

**Conceptualization:** Berhe Etsay Tesfay, Biserka Pop-stefanija, Patrick Keating.

**Data curation:** Berhe Etsay Tesfay.

**Formal analysis:** Berhe Etsay Tesfay, Patrick Keating.

**Investigation:** Berhe Etsay Tesfay.

**Methodology:** Berhe Etsay Tesfay, Patrick Keating.

**Project administration:** Berhe Etsay Tesfay, Abdul Wasay Mullahzada, Samreen Hussain, Monika Slosarska.

**Resources:** Berhe Etsay Tesfay, Destaw Gobezie.

**Software:** Berhe Etsay Tesfay, Patrick Keating.

**Supervision:** Berhe Etsay Tesfay, Destaw Gobezie, Rosita de Boer.

**Validation:** Berhe Etsay Tesfay, Biserka Pop-stefanija, Patrick Keating.

**Visualization:** Berhe Etsay Tesfay.

**Writing – original draft:** Berhe Etsay Tesfay, Patrick Keating.

**Writing – review & editing:** Destaw Gobezie, Ivan Andreas Sinaga, Amanya Jacob, Abdul Wasay Mullahzada, Samreen Hussain, Patrick Keating.

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
