## [Decision Letter · Decision Letter 0]

14 Feb 2024

PONE-D-23-36229Lot quality assurance sampling survey for water, sanitation and hygiene monitoring and evidence-based advocacy in Bentiu IDP camp, South SudanPLOS ONE

Dear Dr. Tesfay,

Thank you for submitting your manuscript to PLOS ONE. After careful consideration, we feel that it has merit but does not fully meet PLOS ONE’s publication criteria as it currently stands. Therefore, we invite you to submit a revised version of the manuscript that addresses the points raised during the review process.

We look forward to receiving your revised manuscript.

Kind regards,

Olushayo Oluseun Olu

Academic Editor

PLOS ONE

Journal Requirements:

3. Please include a complete copy of PLOS’ questionnaire on inclusivity in global research in your revised manuscript. Our policy for research in this area aims to improve transparency in the reporting of research performed outside of researchers’ own country or community. The policy applies to researchers who have travelled to a different country to conduct research, research with Indigenous populations or their lands, and research on cultural artefacts. The questionnaire can also be requested at the journal’s discretion for any other submissions, even if these conditions are not met.  Please find more information on the policy and a link to download a blank copy of the questionnaire here: https://journals.plos.org/plosone/s/best-practices-in-research-reporting. Please upload a completed version of your questionnaire as Supporting Information when you resubmit your manuscript.

Reviewers' comments:

Reviewer's Responses to Questions

**Comments to the Author**

1. Is the manuscript technically sound, and do the data support the conclusions?

Reviewer #1: Yes

Reviewer #2: Partly

2. Has the statistical analysis been performed appropriately and rigorously? 

Reviewer #1: I Don't Know

Reviewer #2: No

3. Have the authors made all data underlying the findings in their manuscript fully available?

Reviewer #1: Yes

Reviewer #2: Yes

4. Is the manuscript presented in an intelligible fashion and written in standard English?

Reviewer #1: Yes

Reviewer #2: Yes

5. Review Comments to the Author

Reviewer #1: The paper is well articulated and mainly descriptive, however the conclusion that if the WASH indicators improve will lead to an interruption in the Hep E outbreak is presumptive. Also South Sudan is one of two countries that have used the Hep E vaccine, a statement on that and its impact would have been helpful. Can you look at the conclusion again and recommendation based on my comments please.

Reviewer #2: MAJOR COMMENT

1)LINE 71-73

In 2016, inadequate WASH contributed to 60% of diarrheal deaths which could have been prevented by improving water and sanitation services [4]. Furthermore, improved hygiene, sanitation, and safe water access can also reduce neglected tropical diseases (such as schistosomiasis and Guinea worm diseases) morbidities by almost 80% [5].

A comprehensive paragraph introducing WASH (full definition before acronym )and its implication is important in the study background. This should be include a short description of the WASH initiative.

2)LINE 198

Nearly all households reported they eat from a shared plate

Is eating from a shared plate an unhygienic indicator? How was this indicator chosen? Any references ? … Have you also considered that shared eating and eating together is a cultural, community and family practice of Sudan? Please explain in detail how the above details were handled-cultural practice versus hygiene. If not kindly find a better indicator to assess hygiene.

3)LINE 176

Table 1. Socio-demographic characteristics of respondents

These table contains too few (not enough) variables to describe the study population. Give more information about the data in the samples/population….More variables --age/age group, mean of age, STD, under 5 children can be described too… new borns? toddlers, younger children … age in months etc. .. occupations if any? Level of education?

All these variable help in discussing the result outcome .. did education or occupation impact result. Discuss?

4)Statistical Rigor

No statistically rigor in this study - only descriptives /summary statistics… do more inferential statistics … compare variables in different scenarios and discuss more on this in the discussion section.

LINE 12-105

In this study, we used the LQAS methodology to quantify the WASH gaps in Bentiu IDP camp, identify the priority areas for intervention, mobilize resources for action, and support advocacy efforts.

The study did not show or discuss how resources were mobilized for action….Rephrase to read : identify the priority areas for intervention, identify areas for resources allocation , and support advocacy efforts

MINOR COMMENTS

5)LINE 96

In the UNHCR WASH in emergency handbook and Sphere guideline, annual knowledge, attitude, and practice

Define UNHCR - in full before utilizing acronym

6. PLOS authors have the option to publish the peer review history of their article (what does this mean?). If published, this will include your full peer review and any attached files.

Reviewer #1: **Yes: **Sylvester Maleghemi

Reviewer #2: No

---

## [Author Response · Author response to Decision Letter 0]

28 Mar 2024

Dear reviewers, 

We sincerely thank you for taking your time to review the manuscript and provide your valuable feedbacks. We have gone through the comments line by line and we have addressed them in the manuscript document when there is a need to change or otherwise we have clarified in the response to reviewers letter. 

As mentioned, while we have taken most of the comments into consideration and addressed them in the manuscript documents, like we have added the WASH background, we added additional relevant sociodemographic, amended the resource mobilisation paragraph for clarity and addressed all the other suggestions. 

LQAS methods are specialised methods designed for access or service coverage monitoring and evaluation they are less precise for statistical inference. In this study our objective was to identify the gaps and prioritise for intervention and hence we used LQAS method. We have clarified the details of this in the manuscript and in the response to reviewers letter. We hope the responses provided to be satisfactory.

Thank you again for your valuable time.

Authors

---

## [Editor Report · Decision Letter 1]

9 Apr 2024

Lot quality assurance sampling survey for water, sanitation and hygiene monitoring and evidence-based advocacy in Bentiu IDP camp, South Sudan

PONE-D-23-36229R1

Dear Dr. Tesfay,

We’re pleased to inform you that your manuscript has been judged scientifically suitable for publication and will be formally accepted for publication once it meets all outstanding technical requirements.

Kind regards,

Olushayo Oluseun Olu

Academic Editor

PLOS ONE
---

## [Editor Report · Acceptance letter]

3 Jun 2024

PONE-D-23-36229R1 

PLOS ONE

Dear Dr. Tesfay, 

I'm pleased to inform you that your manuscript has been deemed suitable for publication in PLOS ONE. Congratulations! Your manuscript is now being handed over to our production team.

Kind regards, 

on behalf of

Dr. Olushayo Oluseun Olu 

Academic Editor

PLOS ONE